# Feed Supplemented with *Aronia melanocarpa* (AM) Relieves the Oxidative Stress Caused by Ovulation in Peak Laying Hens and Increases the Content of Yolk Precursors

**DOI:** 10.3390/ani12243574

**Published:** 2022-12-17

**Authors:** Bo Jing, Huanwei Xiao, Haixu Yin, Yingbo Wei, Haoyuan Wu, Dongliang Zhang, Ivan Stève Nguepi Tsopmejio, Hongmei Shang, Zhouyu Jin, Hui Song

**Affiliations:** 1School of Life Science, Jilin Agricultural University, Changchun 130118, China; 2Engineering Research Center of the Chinese Ministry of Education for Bioreactor and Pharmaceutical Development, Changchun 130118, China; 3College of Animal Science and Technology, Jilin Agricultural University, Changchun 130118, China; 4Engineering Research Center of Chinese Ministry of Education for Edible and Medicinal Fungi, Changchun 130118, China

**Keywords:** *Aronia melanocarpa*, peak laying hens, oxidative stress, egg quality, yolk precursor

## Abstract

**Simple Summary:**

The current work evaluated the protective effect of *Aronia melanocarpa*(AM ) on the liver and ovary of laying hens during the peak laying period and the related mechanisms. In this study, we observed the morphology and histology of liver and ovary and found that AM can alleviate the damage caused by oxidative stress. Serum biochemical parameters and expression abundance of key gene proteins were measured at 33 weeks of age. AM can activate the kelch-like ECH-associated protein 1 (Keap1)/ nuclear factor erythroid-2 related factor 2 (Nrf2) signal pathway, increase the content of antioxidant enzymes, increasing the hormone content, thereby improving the yolk synthesis capacity, and ultimately improving the production performance and egg quality of peak laying hens.

**Abstract:**

The continuous ovulation of laying hens during the peak period is likely to cause oxidative stress, resulting in a reduction in the laying cycle over time. The aim of this study was to evaluate the antioxidant effects of *Aronia melanocarpa* (AM) in the diet and its effect on the yolk precursor content caused by ovulation in laying hens during the peak period. A total of 300 25-week-old Roman brown laying hens were randomly divided into five groups with six replicates in each group, 10 in each replicate. The control group was fed a basal diet, the positive control group was fed a Vitamin C (VC) plus basal diet, and the experimental group was fed 1%, 4%, and 7% doses of AM plus diet according to the principle of energy and nitrogen requirements, which lasted eight weeks. At the end of the study, the egg quality, biochemical, and antioxidant markers, as well as mRNA and protein expressions, were evaluated to determine the potential signaling pathways involved. Results showed that the addition of AM to the feed increased the weight of laying hens at the peak of egg production and improved egg quality. The biochemical markers, as well as the antioxidant parameters in the serum, liver, and ovarian tissues, were ameliorated. The gene and protein expression of recombinant kelch-like ECH-associated protein 1 (Keap1) in the liver and ovarian tissues was decreased, while nuclear factor erythroid-2 related factor 2 (Nrf2) and heme oxygenase-1 (HO-1) expression was increased. The feed supplemented with AM also increased the estrogen contents and lipid parameters, as well as the gene and protein expressions related to the yolk precursor. Feed supplemented with AM could improve the egg quality and the oxidative stress caused by the ovulation process of laying hens during the peak egg production period by activating the Keap1/Nrf2 signaling pathway. These results suggest that the feed supplemented with 1% AM and 4% AM can improve egg production in peak laying hens.

## 1. Introduction

Although commercial laying hens reach the maximum egg production performance in their peak period, their service life is short, and the egg production rate is not high throughout the period. Thus, they still have huge breeding and production potential [1]. The main reasons for the decreased egg production rate of laying hens are the reduction in yolk precursor content and the premature aging of ovaries.

One of the main factors contributing to premature ovarian aging is oxidative stress. Oxidative stress is due to the accumulation of reactive oxygen species (ROS) in the organism, whose physiological levels maintain normal functions, such as signal transduction and redox regulation, while high levels of ROS can induce oxidative damage [2]. During normal conditions, there is a complex antioxidant defense system scavenging ROS and maintaining the redox state of different cells. This antioxidant defense system consists of antioxidant enzymes, such as superoxide dismutase (SOD), catalase (CAT), glutathione peroxidase (GPX), and glutathione S-transferase (GSH-ST), as well as biological antioxidants, which include glutathione (GSH) and vitamins C and E [3].

Nuclear factor erythroid 2-related factor 2 (Nrf2) is a well-established key transcription factor regulating antioxidant genes responsible for inducing various cellular defense mechanisms against oxidative stress [4]. Normally, Nrf2 is localized in the cytoplasm and sequestered by its repressor, Kelch-like ECH-associated protein 1 (Keap1). Under oxidative stress conditions, Keap1 changes its conformation so it can no longer bind Nrf2 molecules. In this case, Nrf2 enters the nucleus and recognizes and binds to ARE (antioxidant response element), which play a role in the antioxidant defense system. Nrf2 can, thus, promote the expression of antioxidant enzymes, such as heme oxygenase-1 (HO-1), to prevent apoptosis [5]. However, the expression of Nrf2 and its downstream gene HO-1 in the ovary and liver as a reflection of tissue antioxidant status remains to be elucidated. At present, most of the studies focus on the ovarian aging of laying hens in the later stage of laying and determining methods to alleviate this ovarian aging [6]. However, part of the reason for this may also be because commercial laying hens produce a large amount of ROS during peak ovulation, which disrupts the ROS balance. This causes the liver and ovary of laying hens to become susceptible to oxidative stress and increase senescence, resulting in shorter egg laying cycle. Therefore, it is necessary to study the approaches to alleviate excessive aging of the ovarian function of laying hens during the peak egg production period, reduce the production of oxidative stress, improve the antioxidant capacity of laying hens, and prolong the laying cycle of laying hens.

Recent studies have found that feed supplemented with dietary polyphenolic antioxidants, such as grape seed extract [7], lycopene [8], and tea [9], can protect the ovarian function of laying hens and delay ovarian aging. *Aronia melanocarpa* (AM) is a unique berry plant of the Rosaceae family, native to North America [10]. At present, the studies available on AM have investigated mainly its fruit. Thus, the AM that subsequently appears in this study refers to the fruit. Their fruits are rich in different nutrients, such as sugars, proteins, and polyphenolic compounds, which have been revealed to be the most important. AM is rich in polyphenolic compounds, including anthocyanins and flavonoids, which show various biological activities. Studies have shown that AM has numerous health benefits, such as effectively scavenging free radicals, antioxidant properties, and protecting against oxidative damage in various tissues [11,12].

The purpose of this study was to alleviate the oxidative stress caused by ovulation in peak laying hens through the keap1/Nrf2 signaling pathway and to increase the content of yolk precursors by adding AM in the feed.

## 2. Materials and Methods

### 2.1. Moral Statement

This study was conducted in accordance with the recommendations of the Ministry of Science and Technology of the People’s Republic of China “Guide to the Care and Use of Experimental Animals”, and in accordance with the Jilin Agricultural University (Changchun, China) “Regulations on the Administration of Experimental Animals”. It also complies with ARRIVE2.0 guidelines. The animal experiment program was approved by the Animal Committee of Jilin Agricultural.

### 2.2. Plant Material and Animal Preparation

The fresh fruits of AM were provided by the Mingqian Green Egg Breeding Professional Cooperative in Liaoyuan City, Jilin Province and dried out under sunlight. First, we put fresh AM on the ground to dry naturally for one month. Then, the dried AM is crushed into powder (water drop hammer mill 60 × Type 45, Luquan City, Hebei Province). A portion was taken for direct feed mixing. Other powders were subject to −40 °C cold storage. The 25-week-old Roman brown laying hens with similar body weight and normal development were selected and kept in the Mingqian Green Egg Breeding Professional Cooperative in Liaoyuan City. Before feeding, the animal room was fumigated and sterilized with formaldehyde and potassium permanganate. A total of 300 25-week-old Roman brown laying hens with similar body weight and normal development were selected and randomly divided into 5 groups with 6 replicates in each group, 10 in each replicate. All groups were fed with diet as shown in Appendix A. The control group was fed with basal diet, the positive control group was fed with basal diet supplemented with 0.05% Vitamin C (VC), and the experimental groups were received diet supplemented with different concentrations of AM (1%, 4%, and 7% dried fruit powder of AM). The hens were housed in wire cages (45 × 45 × 50 cm) with 3 hens per cage and kept in 3-tier ladder-type cages in an environmentally controlled house. Each repetition was evenly distributed in each layer. The trial lasted for 8 weeks from August to September 2019. All hens had free access to clean water and feed three times daily at 7:00 am, 11:00 am, and 5:00 pm. The house temperature was maintained at 24 °C throughout the experiment with 17 h of light and 7 h of darkness.

### 2.3. Sample Collection for Blood and Tissues

Blood and tissue samples were collected 8 weeks after the experiment. Six birds were randomly selected, and their blood samples were collected via the wing vein. The blood sample was centrifuged at 3000× *g* for 15 min to separate blood plasma (*n* = 10/treatment), then kept at –20 °C for the subsequent experiment. The hens were euthanized. Six bird livers and ovaries were collected and immediately frozen in liquid nitrogen and subsequently stored at –80 °C for RNA isolation. Body weight (before slaughter), liver and ovary weight, grade follicle weight (F1, F2, F3, F4 and F5) and diameter were measured.

### 2.4. Organ Index and Morphological Observation

After 8 weeks of the experiment, 6 hens were taken from each experimental group, the livers and ovaries were dissected, washed with normal saline, and weighed to calculate the organ index = organ weight/body weight, and the liver and ovarian tissues were placed in more than 4% fixed in a paraformaldehyde solution. Liver and ovarian fragments were then dehydrated in graded ethanol, cleared in xylene, and embedded in paraffin. Embedded tissue samples were sectioned at 5 μm and mounted on glass slides and stained with Hematoxylin and eosin following standard protocols. The morphological observation was performed using an optical microscope (DP80Digital, Olympus, Tokyo, Japan) and Image Pro plus analysis.

### 2.5. Determination of AST and ALT

Liver and serum concentrations of total protein(A045-3), aspartate aminotransferase (AST, C010-1-1), and alanine aminotransferase (ALT, C009-1-1) were determined according to the manufacturer’s instructions and kits (Nanjing Jiancheng Institute of Bioengineering, Nanjing, China).

### 2.6. Determination of Antioxidant Parameters

Liver, ovary, serum, and yolk concentrations of total superoxide dismutase (T-SOD, A001-1), glutathione peroxidase (GPX, A005-1), total antioxidant capacity (T-AOC, A015-1), and malondialdehyde (MDA, A003-1) concentrations were obtained according to manufacturer’s instructions and kit (Nanjing Jiancheng Bioengineering Institute, Nanjing, China) using biochemical methods.

### 2.7. Assays of Estrogen Levels

Serum estradiol (E2, J4341-A) levels were determined using an enzyme-linked immunosorbent assay (ELISA) kit (Jiangsu Jingmei Biotechnology Co., Ltd., Jiangsu, China) following the manufacturer’s instructions.

### 2.8. Assays of Yolk Precursor Content and Lipid Parameters

Serum was collected for measurement of yolk precursor parameters, such as vitellogenin (VTG, JM-00842C1) and very low-density lipoprotein (VLDL, J7303-A) content, and ovaries were collected for the measurement of yolk precursor receptor parameters, such as very low density lipoprotein receptor (VLDLR, JM-7303-A) content, using an ELISA kit according to the manufacturer’s instructions (Jiangsu Jingmei Biotechnology Co., Ltd., Jiangsu, China). Lipid parameters, triglyceride (TG, A110-1-1) and cholesterol (TC, A111-1-1) contents in serum, liver, ovary, and egg yolk, were determined using kits following the manufacturer’s instructions (Nanjing Jiancheng Institute of Bioengineering, Nanjing, China).

### 2.9. RNA Isolation and Quantitative Real-Time PCR (qRT-PCR)

A part of the liver and ovary were immediately removed and stored at −80 °C. Total RNA was then extracted by the Trizol method. The quality and concentration were determined by agarose gel electrophoresis and nucleic acid quantification, respectively. The latter involved using a nucleic acid quantitative analyzer (Smart Spec Plus BIO-RAD, Hercules, CA, USA). cDNAs were synthesized using Prime Script RT reagent Kit with gDNA Eraser from total RNA and used to quantify gene expression levels with PCR amplification kit and genes specific primers (Appendix A) for *Keap1, Nrf2, HO-1, SOD1, apoB, apoVLDLII*, and *VTGII* in liver tissue samples, as well as *Keap1, Nrf2, HO-1, SOD1, ER-α,* and *VLDLR* in ovarian tissue samples. PCR amplification was performed in triplicate, and the relative quantification of gene expression was analyzed according to the 2^−ΔΔCt^ method.

### 2.10. Protein Extraction and Western Blot Analysis

Total protein was extracted from ovarian and liver lysates from each hen by T-PER tissue protein extraction reagent (Thermo Pierce, novobiotec, Bejing, China) containing a protease inhibitor cocktail (Thermo Pierce, novobiotec, Bejing, China). Protein concentration was assessed by BCA protein assay. Ten microlitres of each sample at the same concentration were added to the wells of SDS-acrylamide gels of different concentrations. All proteins were separated and transferred to PVDF membranes using special equipment. The membrane was incubated with a specific primary antibody for 24 h at 4 °C. After the incubation, each membrane was incubated with the corresponding anti-mouse IgG antibody or anti-rabbit IgG antibody for 1 h. Specific bands were visualized using Spark ECL Western blotting substrate and quantified using ImageJ software. The following primary and secondary antibodies: goat anti- rabbit Keap1, Nrf2 and HO-1 (CST, Boston, MA, USA; 1:1000), goat anti-rabbit secondary antibodies (ptm-bio, Hangzhou, China; 1:2000), and β-actin (ptm-bio, Hangzhou, China; 1:2000) were used as a reference.

### 2.11. Production Performance and Egg Quality Determination

During the experiment, the weekly body weight, egg production, egg production rate, egg weight, and feed intake of each group were recorded, and the feed to egg ratio and average daily feed intake (ADFI) were calculated. After 8 weeks of the experiment, 15 eggs were randomly selected from each treatment, and the egg weight, egg shape index, egg white height, egg yolk color, Hastelloy unit, egg shell strength, and egg shell thickness were measured, respectively. The egg shape index (the long diameter/short diameter of the egg) was measured using a vernier caliper; the eggshell strength was measured using an eggshell strength tester (Model EFG-050, Robotmation company, Tokyo, Japan); an eggshell thickness measuring instrument (Model ETG-1061, Robotmation company, Japan) to detect egg shell thickness; vernier calipers were used to detect egg shape index; an electronic balance was used to measure egg weight; a colorimetric card was used to measure egg yolk color; and a multifunctional egg quality detector (EMT-5200, Robotmation company, Tokyo, Japan) was used to detect Hastelloy units and protein height.

### 2.12. Statistical Analysis

All experiments were repeated at least three times. Data were analyzed by one-way ANOVA with post hoc Dunnett test and independent samples t-test by using SPSS 26.0 software (SPSS, Chicago, IL, USA) and the results were presented as means ± standard deviation (SD). They were considered statistically significant at *p* < 0.05 and *p* < 0.01.

## 3. Results

### 3.1. Effects of Feed Supplemented with AM on Organ Indices of Laying Hens

The feed supplemented with AM, as well as with VC, ameliorated the liver and ovarian indices of laying hens compared with the control group (Table 1).

### 3.2. Effects of Feed Supplemented with AM on Histomorphology during Peak Egg Production

In this experiment, the histological analysis of the liver and ovary was performed. There were no significant differences in the density of hepatic cells, size of the nucleus, and diameter of the intima of the small yellow follicle in the groups receiving a diet supplemented with AM or VC compared to the control group (Figure 1).

### 3.3. Effects of Feed Supplemented with AM on AST and ALT

As shown in Figure 2, there were no significant differences in the liver AST and ALT content in all groups, indicating that the liver was unaffected during the peak egg production period. However, the AST and ALT contents in the serum were significantly decreased in the groups fed with a diet supplemented with VC, 1% AM, and 4% AM in comparison with the control group, while there was no significant difference in the 7% AM group.

### 3.4. Effects of Feed Supplemented with AM on Antioxidant Parameters

The effects of AM on the oxidative stress parameters during the peak egg production period are shown in Table 2. Compared to the group receiving the basal diet, the MDA and T-AOC content in the liver was reduced, while T-SOD levels increased in the AM and VC groups. However, the liver GPX content decreased and increased significantly in the 7% AM and 1% AM groups, respectively, compared to the control group. The ovary T-SOD levels decreased in the group receiving feed supplemented with AM (significant in the 7% AM group), while T-AOC content increased (significant in the 4% AM and 7% AM groups) compared to the control group. On the other hand, there was no significant difference in GPX and MDA content in the ovary in the AM groups compared to the control group. Concerning the antioxidant parameters in the serum, the contents of T-SOD, GPX, and T-AOC increased significantly in the groups fed with AM supplements, while MDA content reduced significantly in the 4% AM and 7% AM groups compared to the control group. Finally, the yolk T-SOD and GPX content increased significantly in the AM and VC groups, while MDA content was significantly decreased in comparison to the control group. However, there was no significant modification in the yolk T-AOC content.

### 3.5. Effects of Feed Supplemented with AM on Keap1/Nrf2 Pathway

The feed supplemented with AM significantly increased the gene and protein expression of Nrf2, HO-1, and SOD1, as well as significantly decreased the gene and protein expression of Keap1 in the liver compared to the control group (Figure 3A–H). Moreover, in the ovary, the diet supplemented with AM significantly increased the gene and protein expression of Nrf2 and HO-1, while it significantly reduced the gene expression of *Keap1* compared to the control group. Interestingly, there was no significant difference in the Keap1 protein (Figure 3I–P). There was no significant change in the gene expression of *SOD1*.

### 3.6. Effects of Feed Supplemented with AM on Estrogen Levels and Their Receptors during Peak Laying Period

In this experiment, the estradiol content (E2) in the serum was significantly elevated in the groups fed with AM supplements compared to the control group (Figure 4A). In contrast, the gene expression of their receptor (*ER-α*) in the liver increased and decreased significantly in the 4% AM and 7% AM groups, respectively, compared to the control group (Figure 4B).

### 3.7. Effects of Feed Supplemented with AM on Lipid Parameters

Triglycerides (TG) and cholesterol (TC) are not only the raw materials for yolk precursors and estrogens, but also for egg yolks. Therefore, when their content increases, the content of yolk precursors and estrogens may also increase. In this experiment, the TG content in the serum was significantly increased in the groups receiving feed supplemented with AM compared to the control group, except for the 7% AM and VC group, wherein the increase was not significant in the liver and ovary (Figure 5A–C). In terms of TC content, there was a significant increase in the liver in all the AM groups in comparison to the control group, while only the 1% AM and 4% AM groups significantly increased the serum TC content, respectively (Figure 5D,E). In the ovary, the 4% AM groups significantly increased the TC content compared to the control group (Figure 5F). In the egg yolk, the 1% AM and 4% AM groups significantly increased the TC content compared to the control group (Figure 5G).

### 3.8. Effects of Feed Supplemented with AM on Gene and Protein Expression of Yolk Precursor and Its Receptor

The content of yolk precursors can affect the laying rate of eggs. As shown in Figure 6, the gene expression of *VTGII* was significantly decreased in the 7% AM group, while those of *apoB* and *apoVLDLII* were significantly increased in the 1% AM and 7% AM groups compared to the control group (Figure 6A–C). The protein expression of VLDL was significantly increased in all the AM groups. However, only the 4% AM group significantly increased the protein expression of VTG, while the 7% AM and VC groups significantly reduced its expression compared to the control group (Figure 6D,E). The gene expression of VLDLR, as well as its protein expression, were significantly increased only in the group fed with 4% AM supplements compared to the control group (Figure 6F,G).

### 3.9. Effects of Feed Supplemented with AM on Changes in Yolk Deposition during Peak Laying Period

The diameter and total weight of graded follicles are intuitive indicators of yolk deposition. As shown in Table 3, feed supplemented with AM significantly increased the F1–F2 grade follicle diameter compared to the control group. Except for the 7% AM group that showed a decrease, the follicle diameter of other grades (F3–F5) and their total weight increased but not significantly compared to the control group.

### 3.10. Effects of Feed Supplemented with AM on Production Performance and Egg Quality of Peak Laying Hens

The content of yolk precursors can affect the egg production rate. From the results of the laying rate of laying hens supplemented with AM for eight weeks (Figure 7), it could be seen that in the first two weeks, the AM and VC groups had a lower egg production rate, but with time, the egg production rate decreased to the lowest in all groups at the fifth week compared with the control. This might have occurred because of the oxidative stress caused by the continuous The AM and VC groups had a smaller decline in laying rate, and with the passage of time, the laying rate of laying hens increased gradually, and the AM and VC groups had a lower laying rate than the control group. This indicates that the addition of AM to the feed can alleviate the oxidative stress response caused by the ovulation of laying hens during the peak laying period and prolong the laying cycle. Production performance and egg quality are intuitive reflections of the lack of oxidative stress in laying hens and egg. The effects of AM on production performance and egg quality of laying hens are shown in Table 4. The results showed that the 1% AM group increased the body weight of the laying hens compared with the control group. There was no significant difference in other groups. The 7% AM group decreased the egg weight of the laying hens compared with the control group. However, there were no significant modifications on other parameters, except for the 4% AM group that reduced the daily feed intake (ADFI) and the feed-to-egg ratio of the laying hens. Therefore, feed supplemented with AM could improve the growth performance of the laying hens. The egg shape index, egg yolk, eggshell strength, protein height, and Hastelloy unit were improved in the AM and VC groups compared with the control group, even though they were not significant.

## 4. Discussion

Poultry is one of the main sources of animal protein for humans. To meet market demand, a variety of high-yielding laying hens have been selected. However, continuous ovulation of laying hens during the peak egg production period leads to an ovarian recession in laying hens, resulting in a sharp egg production rate and a decline. This also limits the useful life of laying hens and causes a reduction of their commercial value. Ovulation leads to an inflammation-like reaction accompanied by the formation of a large amount of ROS. In the ovaries of high-yielding laying hens, the ovulation process that occurs almost every day accelerates the accumulation of ROS in the ovarian tissue of laying hens, thereby causing oxidative stress in tissues [13]. Tissue sections can help directly observe whether the liver and ovarian tissues of laying hens have been subjected to severe oxidative damage. Furthermore, AST and ALT, as markers of oxidative damage, can indicate whether oxidative stress has been caused in the laying hens. The results of HE staining in this study showed no serious oxidative damage in the liver and ovarian tissues among the groups. Compared with the control group, the addition of 1% AM and 4% AM to the diet reduced the serum AST and ALT content, indicating that adding AM to the diet lowered the oxidative damage of the liver and ovarian tissues of laying hens during peak egg production.

Edible fruits, such as blueberries [14], apples [15], and strawberries [16], contain polyphenolic compounds that effectively protect cells from oxidative stress. The mechanisms of the antioxidant effects of these food-based compounds are through their direct scavenging of free radicals or their indirect outcome of increasing endogenous cellular antioxidant potential, which functions by activating the associated signaling pathways [5]. AM is rich in polyphenolic compounds, and polyphenol is the substance with the highest content in AM. It is also an active ingredient that mainly plays a functional role and antioxidant effect, and the total polyphenol content of AM fruit is significantly higher than those of several other small berries, including blueberries [17], red raspberries [18], red currants [19], strawberries [20], and blackberries [21]. Mężyńska et al. [22] found that AM extracts, mainly polyphenols, can scavenge ROS, inhibit oxidase activity, prevent cadmium-induced oxidative stress in mouse liver, and reduce liver damage. Wang et al. [23] found that adding AM strongly inhibits ankle swelling and modulates oxidative stress state in acute gout rats. A study by Liu et al. [7] showed that the oxidative damage gradually accumulated in the ovarian tissue, the decreased expression of antioxidant genes is a key factor for the decline of ovarian function in aging laying hens, and grape seed proanthocyanidin extract could alleviate D-galactose-induced ovarian aging in laying hens. In addition, grape seed proanthocyanidin extract increased the content of antioxidant index, GPX, in the ovary and serum. Studies have confirmed that GPX in serum is related to the removal of extracellular hydrogen peroxide. In this study, it was found that the addition of AM to the feed reduced the MDA content in the liver, serum, ovary, and egg yolk, increased the content of T-SOD in the liver, serum, and egg yolk, improved the antioxidant ability of T-AOC in the ovary, and significantly increased the GPX content in the serum and egg yolk. This indicated that the continuous ovulation of laying hens during the peak laying period could lead to an increase in the content of hydrogen peroxide in the serum, and the addition of AM could increase the GPX levels in the serum of laying hens during the peak laying period, thereby reducing hydrogen peroxide production.

The key factor regulating the expression of antioxidant genes is Nrf2, which is responsible for inducing various cellular defense mechanisms against oxidative stress. Keap1 is an endogenous inhibitor of Nrf2 itself. It is a polypeptide consisting of 627 amino acid residues and binds to Nrf2 through its own DGR domain. Under physiological conditions, Nrf2 remains in the cytoplasm. However, after oxidative stress, Nrf2 enters the nucleus. Nrf2 promotes the expression of antioxidant enzymes, such as HO-1, and resists apoptosis caused by oxidative stress. The activation of the Keap1/Nrf2 pathway is a protective mechanism for cells to cope with oxidative stress [24]. Tanaka et al. [25] found that TBE and its main polyphenolic compounds alleviated LPS-induced oxidative stress in RAW264.7 macrophages by activating the Keap1/Nrf2 pathway. This study found that, compared with the control group, the addition of AM to the feed significantly increased the gene and protein expression of Nrf2 and HO-1 in the liver and ovary and decreased the gene and protein expression of Keap1, which indicated that adding AM to the feed could activate egg production via Keap1/Nrf2 signaling pathway in the liver and ovary tissues of laying hens during peak period to relieve oxidative stress caused by ovulation.

A study found that when oxidative stress occurred in the liver and ovary of laying hens, the secretion of E2 in laying hens decreased, which reduced the laying hen’s egg production rate [26]. In laying hens with oxidative stress caused by aging, increasing estrogen increases yolk deposition. Liu et al. [27] found that lycopene relieved oxidative stress caused by aging and increased estrogen content. This study found that dietary supplementation of AM improved the expression of E2 and its receptors after alleviating the oxidative stress caused by ovulation, which was consistent with the above results.

The yolk precursor material is usually synthesized in the liver and transported to the follicle through blood vessels. The ability of the liver to synthesize it is generally regulated by estrogen. In oviparous animals, this synthesis of yolk precursor substances in the liver, such as VTG and apoB, is regulated by E2. VTG, once synthesized, passes directly into the blood circulation, while apoB combines with VLDL to form apoVLDL before being transported in the blood [28]. Yolk lipids are mainly synthesized in the liver and transported to the ovary by yolk-targeted specific VLDL and phospholipid-rich lipoprotein (also known as vitellogenin), which are recognized by their receptors (VLDLR) on the membrane and brought into the ovary [29]. They are differentiated in the ovarian cytoplasm into triglycerides and cholesterol, which are the raw materials for the yolk precursor synthesis [30]. Feed supplemented with AM increased the triglyceride and cholesterol content in the liver, serum, and ovaries, as well as the cholesterol content and the levels of estrogen and its receptors in egg yolk. Additionally, the expression of the yolk precursors VTG and VLDL and the gene and protein expression of VTGII, apoB, apoVLDLII, and VLDLR increased.

The yolk precursor is an important indicator affecting egg yolk content, and the diameter and weight of the graded follicle are direct indicators of the content of the yolk precursor. As the content of precursors increases, deposition increases. At the same time, a decrease in the content of yolk precursors leads to a decrease in egg production [31]. The level of egg production and body weight are important indicators for evaluating the growth performance of laying hens. The results of this experiment showed that the addition of AM to the feed increased the body weight and egg production rate of laying hens during the peak egg production period, and there were no significant differences in daily feed intake, egg weight, and feed-to-egg ratio. At the same time, through the analysis of the egg production rate at 25–33 weeks, it was observed that at the age of 29 weeks (five weeks), the laying hens at the peak of egg production caused oxidative stress due to continuous ovulation, which reduced the egg production rate, while the feed supplemented with AM and VC alleviated the oxidative stress caused by ovulation and reduced the egg production rate to a lesser extent. Over time, the egg production rate of the AM and VC groups was higher than that of the control group, which also showed that the addition of feed AM not only relieved oxidative stress, but also improved its antioxidant capacity, thereby increasing its egg production rate. The egg shape index is known to have a great influence on hatchability. This experimental study showed that the AM supplementation to the feed improved the egg shape index. Hastelloy unit can reflect the freshness of eggs, and its level can indicate the consistency of egg protein and the quality of protein [32]. This experimental study showed that, compared with the control group, the addition of AM to the feed increased the Hastelloy unit of eggs, indicating that the freshness of eggs was higher. Eggshell thickness is also an important part of egg quality determination, and eggshell thickness has an extremely significant impact on the eggshell damage rate. Eggshell strength is an important indicator reflecting the breakage resistance of eggs. In this study, the eggshell strength was significantly improved, indicating that the probability of egg breaking could be reduced [33]. The yolk weight is an Important indicator to determine the content of yolk precursors. This study found that, compared with the control group, the addition of AM to the feed increased the yolk weight in the eggs. The above results indicated that the addition of AM to the feed could improve the egg quality of the eggs.

## 5. Conclusions

In conclusion, the present study revealed that feed supplemented with AM could alleviate oxidative stress caused by the ovulation of laying hens during peak egg production by activating the Keap1/Nrf2 signaling pathway, enhancing the antioxidant properties in the liver and ovary, and increasing the expression of estrogens and their receptors. This could increase the expression of yolk precursors, and thus improve the growth performance and egg quality (Figure 8). These results suggest that the feed supplemented with 1% AM and 4% AM can improve egg production in peak laying hens.

## Figures and Tables

**Figure 1 animals-12-03574-f001:**
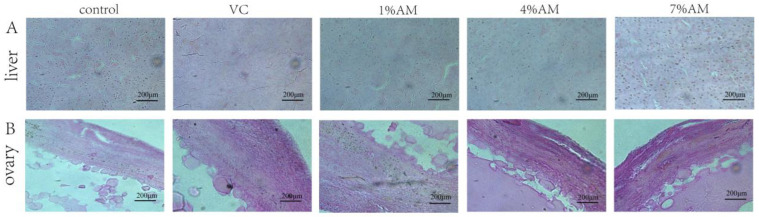
Effects of feed supplemented with AM on liver and ovary histology of laying hens. (**A**) Liver tissue morphology in different groups; (**B**) Ovary tissue morphology in different groups. The bar represents 200 μm.

**Figure 2 animals-12-03574-f002:**
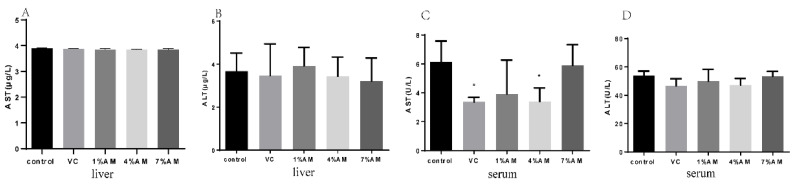
Effects of feed supplemented with AM on AST and ALT of laying hens. Values are shown as mean ± SD (*n* = 6). Significant differences compared to control group are designated as * show significant difference at *p* < 0.05. AST content in liver (**A**), ALT content in liver (**B**), AST content in serum (**C**), ALT content in serum (**D**). Abbreviations: AST, Aspartate aminotransferase; ALT, Alanine aminotransferase.

**Figure 3 animals-12-03574-f003:**
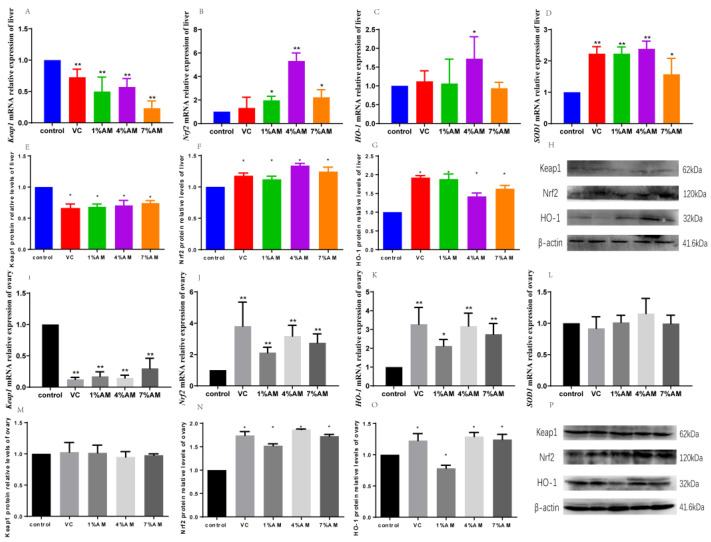
Effects of feed supplemented with AM on Keap1/Nrf2 signaling pathway. Values are shown as mean ± SD (*n* = 6). Significant differences compared to control group are designated as ** show very significant difference at *p* < 0.01, * show significant difference at *p* < 0.05. *Keap1* (**A**), *Nrf2* (**B**), *HO-1* (**C**) and *SOD1* (**D**) gene expressions in liver, Keap1 (**E**), Nrf2 (**F**), HO-1 (**G**) protein expressions in liver, protein gray value in liver (**H**), *Keap1* (**I**)*, Nrf2* (**J**)*, HO-1* (**K**) and *SOD1* (**L**) gene expressions in ovary, Keap1 (**M**), Nrf2 (**N**) and HO-1 (**O**) protein expressions in ovary, protein gray value in ovary (**P**). Abbreviations: Keap1, Kelch-like ECH- associated protein l, Nrf2, Nuclear factor-erythroid 2-related factor 2, HO-1, Heme Oxygenase-1, SOD1, Superoxide Dismutase 1.

**Figure 4 animals-12-03574-f004:**
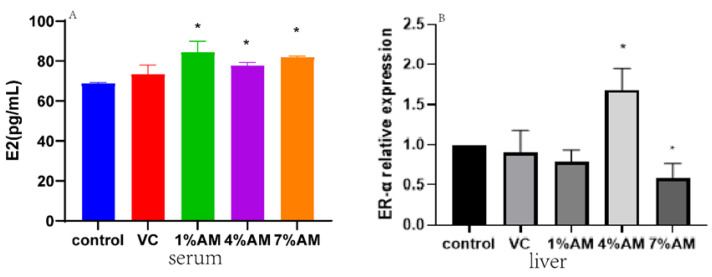
Effects of feed supplemented with AM on E2 and ER-α. E2 content (**A**) in serum, ER-α (**B**) gene expression in liver. Values are shown as mean ± SD (*n* = 6). Significant differences compared to control group are designated as * show significant difference at *p* < 0.05. Abbreviations: E2, Estrogen, *ER-α*, Estrogen receptor-α.

**Figure 5 animals-12-03574-f005:**
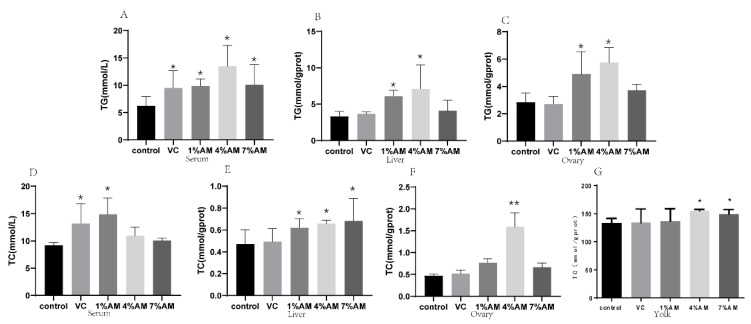
Effects of feed supplemented with AM on TG and TC. TG content in serum (**A**), liver (**B**) and ovary (**C**), TC content in serum (**D**), liver (**E**), ovary (**F**) and yolk (**G**). Values are shown as mean ± SD (*n* = 6). Significant differences compared to control group are designated as ** show very significant difference at *p* < 0.01, * show significant difference at *p* < 0.05. Abbreviations: TG, Triglyceride, TC, Cholesterol.

**Figure 6 animals-12-03574-f006:**
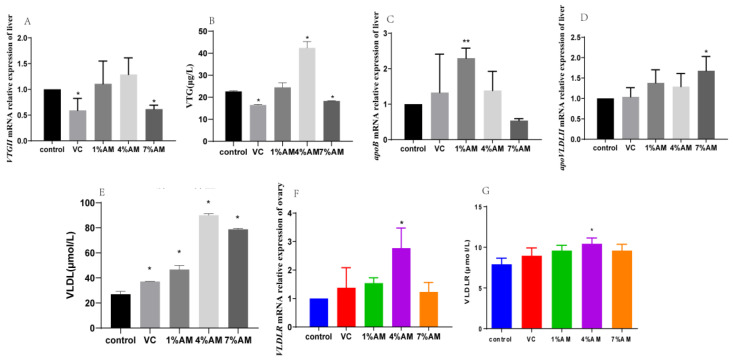
Effects of feed supplemented with AM on yolk precursor and receptor. *VTGII* (**A**), apoB (**C**) and *apoVLDLII* (**D**) gene expressions in liver, VLDLR (**F**) gene expression in ovary, VTG (**B**), VLDL (**E**) and VLDLR (**G**) content in serum. Values are shown as mean ± SD (*n* = 6). Significant differences compared to control group are designated as ** show very significant difference at *p* < 0.01, * show significant difference at *p* < 0.05. Abbreviations: VTG, vitellogenin, VLDL, Very low density lipoprotein, VLDLR, very low density lipoprotein receptor.

**Figure 7 animals-12-03574-f007:**
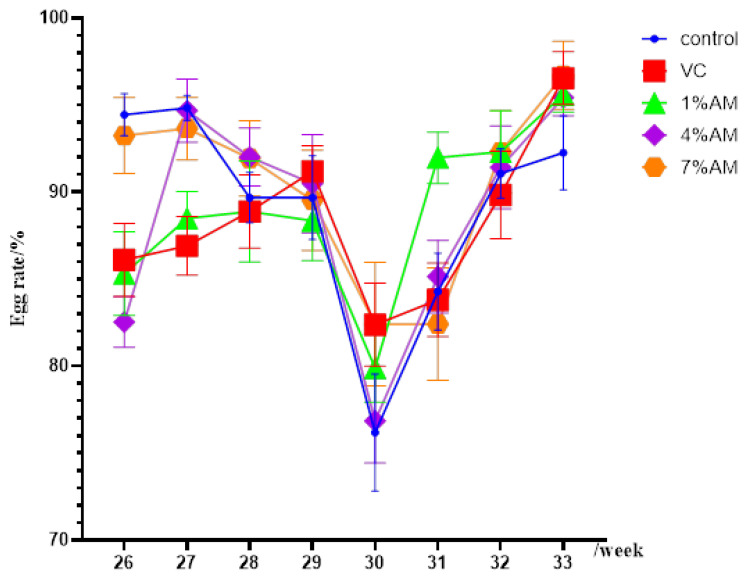
Effect of feed AM supplementation on egg production rate of laying hens at 25–33 weeks during peak laying period.

**Figure 8 animals-12-03574-f008:**
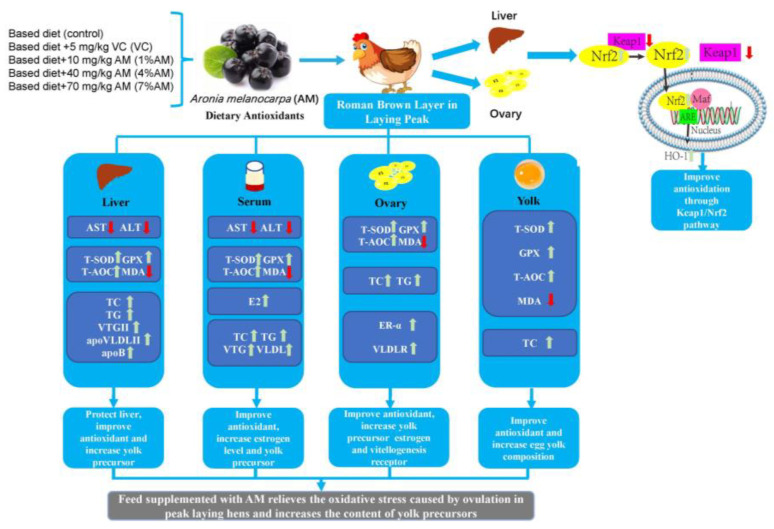
Feed supplemented with AM relieves the oxidative stress caused by ovulation in peak laying hens and increases the content of yolk precursors. **↑** means rise; **↓** means reduce.

**Table 1 animals-12-03574-t001:** Effects of feed supplemented with AM on organ index of laying hens.

Items	Control	VC	1% AM	4% AM	7% AM	*p*-Value
Liver index	27.72 ± 1.48	29.94 ± 1.75	28.70 ± 1.94	29.30 ± 3.52	29.84 ± 1.63	0.4033
Ovary index	49.39 ± 5.16	50.85 ± 5.89	49.43 ± 2.05	49.69 ± 4.23	51.69 ± 3.80	0.8631

Values are shown as mean ± SD (*n* = 6).

**Table 2 animals-12-03574-t002:** Effects of feed supplemented with AM on antioxidant parameters.

Items	Control	VC	1% AM	4% AM	7% AM	*p*-Value
Liver-GPX	132.70 ± 8.00	131.90 ± 5.49	172.00 ± 11.78 **	126.60 ± 0.93	102.60 ± 17.20 *	<0.0001
Liver-MDA	2.13 ± 0.12	1.95 ± 0.18	2.10 ± 0.06	1.54 ± 0.46**	1.77 ± 0.25	0.0023
Liver- T-SOD	13.68 ± 0.87	14.00 ± 0.14	14.25 ± 0.48	14.14 ± 0.22	14.48 ± 0.31	0.0859
Liver- T-AOC	0.87 ± 0.04	0.93 ± 0.02 **	0.94 ± 0.01 **	0.90 ± 0.05	0.88 ± 0.02	0.0024
Ovary-GPX	133.70 ± 19.12	140.40 ± 19.20 *	182.40 ± 10.90 **	225.70 ± 39.00 **	113.70 ± 11.92	<0.0001
Ovary-MDA	1.49 ± 0.34	0.91 ± 0.11*	1.07 ± 0.34	1.19 ± 0.30	1.51 ± 0.32	0.0055
Ovary-T-SOD	8.57 ± 0.81	20.62 ± 6.28 **	16.91 ± 0.92 **	15.83 ± 0.90 **	12.46 ± 0.58 **	<0.0001
Ovary-T-AOC	0.38 ± 0.06	0.44 ± 0.02	0.42 ± 0.05	0.46 ± 0.04 **	0.57 ± 0.01 **	<0.0001
Serum-GPX	38.46 ± 4.00	403.60 ± 61.48 **	676.40 ± 205.10 **	839.20 ± 160.50 **	664.10 ± 220.30 **	<0.0001
Serum-MDA	4.95 ± 0.85	2.31 ± 0.48 **	4.48 ± 0.70	3.81 ± 0.83 *	2.72 ± 0.90 **	<0.0001
Serum-T-SOD	84.78 ± 3.11	111.00 ± 0.80 **	110.90 ± 0.52 **	113.30 ± 1.60 **	111.30 ± 2.80 **	<0.0001
Serum-T-AOC	0.60 ± 0.04	0.59 ± 0.02	0.72 ± 0.02 **	0.66 ± 0.02 *	0.57 ± 0.01	<0.0001
Yolk-GPX	20.53 ± 1.39	102.70 ± 21.67 **	109.30 ± 6.58 **	80.38 ± 5.27 **	81.82 ± 7.28 **	<0.0001
Yolk-MDA	1.52 ± 0.06	0.82 ± 0.07 **	0.43 ± 0.08 **	1.05 ± 0.18 *	0.67 ± 0.08 **	<0.0001
Yolk- T-SOD	82.64 ± 4.01	99.38 ± 3.71 **	92.89 ± 4.33 *	93.83 ± 2.32 **	100.60 ± 3.53 **	<0.0001
Yolk- T-AOC	0.72 ± 0.03	0.71 ± 0.04	0.72 ± 0.06	0.73 ± 0.01	0.73 ± 0.04	0.8950

Values are shown as mean ± SD (*n* = 6). Significant differences compared to control group are designated as ** show very significant difference at *p* < 0.01, * show significant difference at *p* < 0.05. Abbreviations: GPX, Glutathione peroxidase, MDA, Malondialdehyde, T-SOD, Total superoxide dismutase, T-AOC, Total antioxidant capacity.

**Table 3 animals-12-03574-t003:** Diameters and mass of hierarchical of laying hens.

Items	Control	VC	1% AM	4% AM	7% AM	*p*-Value
F1 (cm)	3.03 ± 0.08	3.23 ± 0.12	3.27 ± 0.08 *	3.30 ± 0.29 *	3.07 ± 0.10 *	0.0172
F2 (cm)	2.60 ± 0.06	3.13 ± 0.23 **	2.95 ± 0.14 **	3.02 ± 0.31 **	3.03 ± 0.05 **	0.0006
F3 (cm)	2.33 ± 0.12	2.73 ± 0.48	2.57 ± 0.15	2.70 ± 0.33	2.30 ± 0.31	0.0609
F4 (cm)	2.10 ± 0.13	2.32 ± 0.38	2.10 ± 0.18	2.22 ± 0.37	2.00 ± 0.24	0.3420
F5 (cm)	1.72 ± 0.33	1.93 ± 0.33	1.70 ± 0.23	1.80 ± 0.26	1.53 ± 0.41	0.3083
Hierarchical follicles (F1–F5) mass (g)	44.64 ± 4.88	54.19 ± 6.58 *	47.52 ± 5.07	49.19 ± 4.41	44.75 ± 2.25	0.0128

Values are shown as mean ± SD (*n* = 6). Significant differences compared to control group are designated as ** show significant difference at *p* < 0.01, * show significant difference at *p* < 0.05.

**Table 4 animals-12-03574-t004:** Effects of AM on production performance and egg quality of laying hens.

Items	Control	VC	1% AM	4% AM	7% AM	*p*-Value
**Production Performance**						
BW/kg	1.82 ± 0.03	1.92 ± 0.03	1.95 ± 0.03 *	1.90 ± 0.03	1.89 ± 0.03	0.033
EW/g	62.14 ± 0.88	59.97 ± 0.77	60.86 ± 0.73	60.03 ± 0.60	58.61 ± 0.67 *	0.017
Egg rate %	86.31 ± 1.67	88.14 ± 1.48	89.95 ± 1.42	89.36 ± 2.23	88.46 ± 1.80	0.277
ADFI/g	126.67 ± 3.21	116.70 ± 6.92	148.80 ± 13.92	116.65 ± 12.20	130.49 ± 7.84	0.166
FER	2.21 ± 0.07	2.06 ± 0.11	2.62 ± 0.26	2.06 ± 0.21	2.30 ± 0.13	0.172
**Egg Quality**						
Egg shape index	1.26 ± 0.01	1.25 ± 0.01	1.28 ± 0.01	1.27 ± 0.01	1.26 ± 0.00	0.197
Egg yolk color	9.11 ± 0.14	8.86 ± 0.16	9.42 ± 0.15	8.67 ± 0.16	9.17 ± 0.15	0.007
ETS/mm	0.31 ± 0.01	0.29 ± 0.01	0.31 ± 0.01	0.28 ± 0.01 **	0.29 ± 0.00	0.001
ESH/kg.cm^−2^	54.61 ± 0.58	57.89 ± 1.31	57.95 ± 1.23	55.38 ± 1.04	55.94 ± 1.25	0.037
PH/mm	7.57 ± 0.19	7.39 ± 0.11	7.69 ± 0.17	7.95 ± 0.16	7.59 ± 0.17	0.171
Hastelloy unit	85.86 ± 1.19	85.76 ± 0.62	87.14 ± 0.91	88.15 ± 0.91	87.16 ± 0.94	0.134
Yolk weight/mg	32.00 ± 0.00	32.70 ± 0.00	34.30 ± 0.00	33.80 ± 0.00	32.60 ± 0.00	0.104
EWW/mg	72.10 ± 0.01	72.30 ± 0.01	73.00 ± 0.01	72.10 ± 0.01	70.00 ± 0.01	0.473

Values are shown as mean ± SD (*n* = 6). Significant differences compared to control group are designated as ** show very significant difference at *p* < 0.01, * show significant difference at *p* < 0.05. Abbreviations: BW, Body weight; EW, Egg weight; ADFI, Average daily intake; FER, Feed to egg ratio; ETS, Eggshell thickness; ESH, Eggshell strength; PH, Protein height; EWW, Egg white weight.

## Data Availability

All data included in this study are available upon request by contacting the corresponding author.

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
