# Peer review of "Feed Supplemented with Aronia melanocarpa (AM) Relieves the Oxidative Stress Caused by Ovulation in Peak Laying Hens and Increases the Content of Yolk Precursors"

_animals, 2022, doi:10.3390/ani12243574_

Round 1

Reviewer 1 Report

This manuscript examines feed supplemented with Aronia melanocarpa (AM) relieves the oxidative stress caused by ovulation in peak laying hens and increases the content of yolk precursors. 

This work is interesting, but a few questions are left unanswered 

Please provide a table of what the basal diet is made up of, if possible, the calculates and analyzed after adding the AM to the diet

Can AM be produced in commercial quantities.

Line 22: remove second thereby

Line 28: Do you have to spell out "methods" in the abstract?

Line 28-29: Remove "in this study"

Line 30 replace the comma with "and"

Line 35: Do you have to spell out "results" in the abstract?

Line 36: at what age is peak egg production?

Line 42: Do you have to spell out occlusion in the abstract?

Line 42: remove "The" in front of feed

Line 51: replace service life with egg production phase and remove the "egg" before production

line 55: Cite

Line 95 should it be "peak laying period"?

Line 99: Do you have an IACUC number?

Line 108: Please include what was further done to the AM after drying. Was it ground? what was the form? how was it stored before being applied to the

feed

Line 118: Change receiving to received diets

Line 125 (2.3). You don't have body weight captured in your subtitle.

2.3-2.4: There are some repetitions that can be corrected

Line 312-313: I recommend that you change your statement to improved ------ remove "can"

Double check some of the figures to make sure that they are clear and not blurry 

Author Response

Point 1: Please provide a table of what the basal diet is made up of, if possible, the calculates and analyzed after adding the AM to the diet

Response 1: Thank you very much for your advice. My diet formula is included in the Supplementary table.

Supplementary Table:

Table S1. Composition and nutrient levels of diets.

Item, %

control

VC

 1%AM

4%AM

7%AM

Corn

63.150

63.150

61.930

58.190

54.490

Soybean meal

25.700

25.700

25.860

26.330

26.800

Stone powder

8.930

8.930

8.930

8.930

8.930

Steamed bone meal

0.700

0.700

0.700

0.700

0.700

Soybean oil

0.750

0.750

0.810

1.080

1.310

Salt

0.300

0.300

0.300

0.300

0.300

Choline chloride

0.100

0.100

0.100

0.100

0.100

Vitamin–mineral premixa

1.000

1.000

1.000

1.000

1.000

Methionine

0.120

0.120

0.120

0.120

0.120

Lysine

0.100

0.100

0.100

0.100

0.100

Vitamin C

0.05

Aronia melanocarpa

1

4

7

Total

100.85

100.90

100.85

100.85

100.85

Nutrient level, %

Metabolizable energy

(MC/Kg)

11.30

11.30

11.30

11.30

11.30

Crude protein

17.40

17.40

17.40

17.40

17.40

Ca

3.36

3.36

3.36

3.36

3.36

Lys

0.95

0.95

0.95

0.95

0.95

Met

0.39

0.39

0.39

0.39

0.39

Met+Cys

0.30

0.30

0.30

0.30

0.30

Crude fiber

2.55

2.55

2.58

2.68

2.78

Available phosphorus

0.10

0.10

0.10

0.10

0.10

NaCl

0.29

0.29

0.29

0.29

0.29

Trace elements and vitamin are provided by the Mingqian Green Layer Breeding Professional Cooperative in Liaoyuan City, Jilin Province.

Note: The nutrient content in feed formulation is theoretical calculation values.

a Vitamin-mineral premix provide (per kg diet): 25,300 IU vitamin A, 7,000 IU vitamin D 3, 270 IU vitamin E, 45 mg vitamin K3, 32 mg vitamin B1, 110 mg vitamin B2, 30 mg vitamin B6, 0.6 mg vitamin B12, 40 mg niacin, 13 mg pantothenic acid, 13 mg folic acid, 0.8 mg biotin, 100 mg choline chloride, 60 mg Fe,1.4 mg Cu, 60 mg Mn,13 mg Zn, 0.012 mg Se, 0.098 mg I.

Point 2: Can AM be produced in commercial quantities.

Response 2: The fresh fruits of AM were provided by the Mingqian Green Egg Breeding Pro-fessional Cooperative in Liaoyuan City, Jilin Province. In order to study the influence of AM on laying hens, they specially planted on the mountains around the chicken farm, so AM can be produced in commercial quantities.

Point 3: Line 22: remove second thereby

Line 28: Do you have to spell out "methods" in the abstract?

Line 28-29: Remove "in this study"

Line 30 replace the comma with "and" v 

Line 35: Do you have to spell out "results" in the abstract?

Response 3: Thank you very much for your advice. I have modified my word according to your requirements. (in red)

Point 4: Line 36: at what age is peak egg production?

Line 42: Do you have to spell out occlusion in the abstract?

Line 42: remove "The" in front of feed

Line 51: replace service life with egg production phase and remove the "egg" before production

line 55: Cite

Response 4: Thank you very much for your advice. I have modified my word according to your requirements and The peak egg laying period is divided according to the 2018 edition of the Heilan Brown Layer Raising Management Manual, which says that 23-36 weeks old is the peak egg laying period. (in red)

Point 5: Line 95 should it be "peak laying period"?

Line 99: Do you have an IACUC number?

Response 5: Thank you very much for your advice. I've changed to peak laying hens. I don’t have an IACUC number. But I give editor an exemption declaration. I can provide this later if you need to see it.

Point 6: Line 108: Please include what was further done to the AM after drying. Was it ground? what was the form? how was it stored before being applied to the feed

Response 6: First, we put fresh AM on the ground to dry naturally for one month. Then, the dried AM is crushed into powder (water drop hammer mill 60 × Type 45, Luquan City, Hebei Province). Take a portion for direct feed mixing. Other powders are sent to - 40 ℃ cold storage. I revised the materials and methods of the article again.

Point 7: Line 118: Change receiving to received diets

Line 125 (2.3). You don't have body weight captured in your subtitle.

2.3-2.4: There are some repetitions that can be corrected

Line 312-313: I recommend that you change your statement to improved ------ remove "can"

Double check some of the figures to make sure that they are clear and not blurry

Response 7: hank you very much for your advice. I have modified my word according to your requirements.

Reviewer 2 Report

I have completed my evaluation of this manuscript.

Section  Material and Methods, and Results, and Discussion needs to be corrected.

1.   Line 112: You wrote: ‘hens had free access to feed’ and in line 122 ‘hens had.. feed three times daily at 7:00 am, 11:00 am and 5:00 pm’ It’s not clear. Feed  was ad libitum? Explain this.

2.   Line 145:  and total protein?

3.   Line 146-148: Describe in detail assay methods of  protein, ALT and AST determination, and aparatures used.

4.   Paragraph 2.6-2.8: Describe in detail assay methods and apparatuses used.

5.   Paragraph 2.5-2.6 and 2.8: Describe  assay methods tissue extraction (liver, ovary and yolk)  for analysis.

6.   Line 136-138: This is a replication of a passage paragraph 2.3

7.   Paragraph 2.11: What was the mortality rate or livability?

8.   Paragraph 2.12: The experimental unit of each data should be clearly stated.

9.   Results and Discussion: In your study,  different doses of AM had variable impact on the parameters tested, why did you write most often only AM without specifying the percentage of AM? The authors didn’t analyze the different doses of AM in paragraph Results. So, the authors cannot make a good discussion for this work. You should precisely address your results and interpret them adequately. You must revise it.

10.  Paragraph Results: In this paragraph you should only describe your results. This is not the place for conclusions and discussions (for example line: 215-216, 238-239, 264-266, 279-281, 311-313, 361-362, 364-368, 372-377). Delete these fragments.

11.  Figure 7:  This figure is illegible. Change the scale of the Y axis and make it bigger or you should present dates in the table. Why haven’t  you presented a significant difference? Please revise.

12.  Paragraph 3.10: In my opinion, the use of ‘was improve’ e.t.c for statistically insignificant numerical differences is not allowed. You must revise it. If you choose to comment or discuss non-significant effects, you must be honest and do the same for all response variables, whether the numerical differences are positive or negative. Please delete those formulations and properly revise your discussion.

13.  Table 4:  

·        Egg rate p-value is 0.277. Why do you designate difference for the VC group? 

·       Egg yolk color p-value is 0.007. Why didn’t you designate differences?

·       ETH p-value is 0.037. Why didn’t you designate differences?

·       Abbreviations: You put them in order as in the table.

·       In abbreviations you are missing ETH or is this ESH?

14. In table 4 you should describe the scale or unit of yolk color.

15. Conclusions: for which doses of AM  is true this conclusion?

Author Response

Point 1:  Line 112: You wrote: ‘hens had free access to feed’ and in line 122 ‘hens had.. feed three times daily at 7:00 am, 11:00 am and 5:00 pm’ It’s not clear. Feed  was ad libitum? Explain this.

Response 1: Thank you very much for your advice. All our hens can get clean water and feed for free at 7:00 a.m., 11:00 a.m. and 5:00 p.m. every day. Free diet is really my problem, and I have deleted it from my manuscript.

Point 2: Line 145:  and total protein?

Line 146-148: Describe in detail assay methods of  protein, ALT and AST determination, and aparatures used.

Response 2: Thank you very much for your advice. The detailed methods in the kit have been separately put in a compressed package. Use the 721 spectrophotometer to measure the OD value(Shanghai No. 1 Optical Instrument Factory).

Point 3: Paragraph 2.6-2.8: Describe in detail assay methods and apparatuses used.

Response 3: Thank you very much for your advice. The detailed methods in the kit have been separately put in a compressed package. Paragraph 2.6: Use the 721 spectrophotometer to measure the OD value(Shanghai No. 1 Optical Instrument Factory). Paragraph 2.7-2.8: Use the DNM-9606 enzyme label analyzer to measure the OD value(Beijing Planck New Technology Co., Ltd).

Point 4: Paragraph 2.5-2.6 and 2.8: Describe  assay methods tissue extraction (liver, ovary and yolk)  for analysis.

Response 4: Thank you very much for your advice. I didn't explain the specific method of extracting liver, ovary and egg yolk in my manuscript, because different experimental extracts are different, so I didn't show it in my manuscript.

Point 5: Line 136-138: This is a replication of a passage paragraph 2.3

Response 5: Thank you very much for your advice. Because the formula of organ index of direct radiation is easy to understand, I mentioned the sampling again.

Point 6: Paragraph 2.11: What was the mortality rate or livability?

Response 6: Thank you very much for your questions. Our mortality rate is 0%.

Point 7: Paragraph 2.12: The experimental unit of each data should be clearly stated.

Response 7: Thank you very much for your advice. I have revised it.

Point 8: Results and Discussion: In your study, different doses of AM had variable impact on the parameters tested, why did you write most often only AM without specifying the percentage of AM? The authors didn’t analyze the different doses of AM in paragraph Results. So, the authors cannot make a good discussion for this work. You should precisely address your results and interpret them adequately. You must revise it.

Response 8: Thank you very much for your advice. I wrote AM to explain that the effect of all AM groups is obviously different from that of the control group. There are some data that the effect of individual groups is better than that of other groups, and I have revised them again.

Point 9: Paragraph Results: In this paragraph you should only describe your results. This is not the place for conclusions and discussions (for example line: 215-216, 238-239, 264-266, 279-281, 311-313, 361-362, 364-368, 372-377). Delete these fragments.

Response 9: Thank you very much for your advice. I have delete these fragments.

Point 10: Figure 7: This figure is illegible. Change the scale of the Y axis and make it bigger or you should present dates in the table. Why haven’t you presented a significant difference? Please revise.

Response 10: Thank you very much for your advice. I have made changes according to your suggestions.

Point 11: Paragraph 3.10: In my opinion, the use of ‘was improve’ e.t.c for statistically insignificant numerical differences is not allowed. You must revise it. If you choose to comment or discuss non-significant effects, you must be honest and do the same for all response variables, whether the numerical differences are positive or negative. Please delete those formulations and properly revise your discussion.

Response 11: Thank you very much for your advice. I have made changes according to your suggestions.

Point 12: Table 4: 

Egg rate p-value is 0.277. Why do you designate difference for the VC group?

Egg yolk color p-value is 0.007. Why didn’t you designate differences?

ETH p-value is 0.037. Why didn’t you designate differences?

Abbreviations: You put them in order as in the table.

In abbreviations you are missing ETH or is this ESH?

Response 12: Thank you very much for your questions.“Egg yolk color p-value is 0.007 and ETH p-value is 0.037”. I didn’t designate differences . This is because there is no significant difference compared with the control group, but there is a significant difference between the 1% AM group and the 4% AM group. Therefore, I say there is no significant difference compared with the control group. ETH is that I made a mistake and I revised it again.

Point 13: In table 4 you should describe the scale or unit of yolk color.

Conclusions: for which doses of AM  is true this conclusion?

Response 13: Thank you very much for your advice. We use Roche color comparison fan (15 grades) to detect the yolk color, but no unit is found, so I have not written any unit here. Conclusion I have revised it here.